# Cell Type-Specific Extracellular Vesicles and Their Impact on Health and Disease

**DOI:** 10.3390/ijms25052730

**Published:** 2024-02-27

**Authors:** Sohil Amin, Hamed Massoumi, Deepshikha Tewari, Arnab Roy, Madhurima Chaudhuri, Cedra Jazayerli, Abhi Krishan, Mannat Singh, Mohammad Soleimani, Emine E. Karaca, Arash Mirzaei, Victor H. Guaiquil, Mark I. Rosenblatt, Ali R. Djalilian, Elmira Jalilian

**Affiliations:** 1Department of Ophthalmology and Visual Sciences, Illinois Eye and Ear Infirmary, University of Illinois at Chicago, Chicago, IL 60612, USA; sohila2@uic.edu (S.A.); hmasso2@uic.edu (H.M.); dtewari@uic.edu (D.T.); aroy42@illinois.edu (A.R.); mchaud46@uic.edu (M.C.); cjaza2@uic.edu (C.J.); akris4@uic.edu (A.K.); msing7@uic.edu (M.S.); msolei2@uic.edu (M.S.); dremineesra@gmail.com (E.E.K.); vguaiqui@uic.edu (V.H.G.); mrosenbl@uic.edu (M.I.R.); jalilian@uic.edu (E.J.); 2Richard and Loan Hill Department of Bioengineering, University of Illinois at Chicago, Chicago, IL 60607, USA; 3Department of Ophthalmology, University of Health Sciences, Ankara Bilkent City Hospital, Ankara 06800, Turkey; 4Department of Ophthalmology, University of Medical Sciences, Farabi Eye Hospital, Tehran 13366 16351, Iran; drarashmirzaei@gmail.com

**Keywords:** cell type-specific extracellular vesicles, extracellular vesicles, exosomes, intercellular communication, diagnostic biomarkers, disease pathogenesis

## Abstract

Extracellular vesicles (EVs), a diverse group of cell-derived exocytosed particles, are pivotal in mediating intercellular communication due to their ability to selectively transfer biomolecules to specific cell types. EVs, composed of proteins, nucleic acids, and lipids, are taken up by cells to affect a variety of signaling cascades. Research in the field has primarily focused on stem cell-derived EVs, with a particular focus on mesenchymal stem cells, for their potential therapeutic benefits. Recently, tissue-specific EVs or cell type-specific extracellular vesicles (CTS-EVs), have garnered attention for their unique biogenesis and molecular composition because they enable highly targeted cell-specific communication. Various studies have outlined the roles that CTS-EVs play in the signaling for physiological function and the maintenance of homeostasis, including immune modulation, tissue regeneration, and organ development. These properties are also exploited for disease propagation, such as in cancer, neurological disorders, infectious diseases, autoimmune conditions, and more. The insights gained from analyzing CTS-EVs in different biological roles not only enhance our understanding of intercellular signaling and disease pathogenesis but also open new avenues for innovative diagnostic biomarkers and therapeutic targets for a wide spectrum of medical conditions. This review comprehensively outlines the current understanding of CTS-EV origins, function within normal physiology, and implications in diseased states.

## 1. Introduction 

Extracellular vesicles (EVs) are a diverse group of lipid bilayer-encased particles that are characterized by their heterogeneity in size (ranging from 30 to 1000 nm), origin (secreted by nearly all cell types), and composition [1,2]. These endogenously produced particles have the inherent capacity to convey a wide range of bioactive molecules comprising proteins, nucleic acids, and lipids. EV cargo is absorbed and used to elicit a cellular signaling response and, thus, can play a crucial role in intra- and intercellular communications, angiogenesis, cancer progression, homeostasis, and immune and inflammatory responses [3,4,5,6,7]. There are multiple classifications of EVs based on their origin, size, and functionality. Three major sub-types of EVs are apoptotic bodies (1000–5000 nm), microvesicles (MVs) or ectosomes (100–1500 nm), and exosomes (30–100 nm) [1,8,9,10]. EVs are found in all body fluids, including plasma, urine, breast milk, saliva, and tears [11]. Their widely available biodistribution not only signifies their role in normal physiological activities but also their potential to affect pathological states [12]. The field of EV detection, purification, and analysis is still in its infancy, and the best techniques are still being debated [13]. 

The field of EVs is gaining great attention because of EVs’ unique role in intercellular communication. Predominant research in this field has concentrated on the therapeutic applications of mesenchymal stem cell-derived EVs (MSC-EVs), which have demonstrated a significant effect across diverse diseases [14,15], including neurological disorders [3], cardiovascular disorders [16], respiratory disorders [17], metabolic disorders [18], ophthalmic conditions [1,19], and various cancers [20,21]. For example, utilizing their ability to cross the blood–brain barrier, MSC-EVs have been employed for the encapsulation of curcumin and as a magnetic resonance imaging contrast agent in the treatment of Parkinson’s disease (PD), with the miRNA within these exosomes, additionally demonstrating a preventive effect on neuronal death and a reduction in PD symptoms [22,23].

However, more recently, cell type-specific EVs (CTS-Evs) have also begun to be recognized as pivotal entities in the realm of intercellular communication, contributing significantly to our understanding of cellular interactions in both physiological and pathological contexts. CTS-Evs are essential for orchestrating a range of homeostatic cellular activities, including growth, proliferation, and differentiation, and their composition changes in response to cellular stress and pathological status. Their tissue-specific origins allow them to transport specific biomolecules from their parent cells and enable them to selectively influence target cells and the surrounding microenvironment, thereby facilitating specific cellular functions and signaling pathways critical for maintaining homeostasis, differentiation, immune modulation, and tissue regeneration [24]. Additionally, this specificity and ability to be extracted from biofluids enables CTS-EVs to serve as valuable biomarkers of disease [25]. The dysregulation of CTS-EV-mediated intercellular communication has been implicated in various diseases, including cancer, neurological disorders, autoimmune conditions, cardiovascular diseases, and more (Figure 1) [26]. Therefore, understanding the mechanisms and functions of CTS-EVs can provide insights into disease pathogenesis and potentially lead to the identification of both diagnostic and therapeutic targets. In the present review, we will highlight the role of CTS-EVs in both physiological and pathological conditions.

## 2. EV Origins and Intercellular Communication

### 2.1. Biogenesis and Release

CTS-EVs are produced through intricate biogenesis pathways unique to their cell of origin. These vesicles originate from endosomal compartments or plasma membranes, depending on the type of EV [27]. Apoptotic bodies are the largest EVs, roughly 1 to 5 μm in diameter, making them in the size range of smaller cells [28]. Apoptotic bodies, as the name implies, are only formed during programmed cell death [29]. Microvesicles (MVs) are a subtype of EVs that range in size from 100 to 1000 nm [9,10]. They are formed and released via the outward budding and fission of the plasma membrane (blebbing), though their role in intercellular communication is less well-defined compared with other EV classifications [27]. Exosomes, with a size in the 30–150 nm range [3,30], are formed within endosomal compartments. The process begins via endocytosis of the cell’s plasma membrane with internalized cargoes to form an early endosome [31]. These early endosomes mature into multivesicular bodies (MVBs), rich with many intraluminal vesicles (ILVs). ILV formation relies on reorganization via tetraspanins and recruitment via endosomal sorting complexes required for transport (ESCRTs) [27,32,33]. These ILVs contain a variety of cargo loaded from the Golgi apparatus. The MVB is then transported to the plasma membrane along the cytoskeletal and microtubule network. Once the MVB docks with the plasma membrane, the exocytosis process occurs, and the various intraluminal vesicles, now called exosomes, are released into the extracellular space [31]. They either remain in the extracellular space or release throughout wider vasculature, with widely available biodistribution being observed in most organs [34]. It is important to note that a vast majority of EV-based applications in the study of mechanisms, diagnostics, and therapeutics refer to cell type-specific exosomes, in particular, isolated through centrifugation techniques, so there are not as many applications for the other types of extracellular vesicles, such as microvesicles and apoptotic bodies.

### 2.2. Molecular Composition

The content of CTS-EVs varies with respect to their cell type, physiological condition, and disease status [27]. Proteomic and genomic analyses have revealed that CTS-EVs contain a wide array of bioactive molecules, including a number of cell-specific nucleic acids, proteins, lipids, and glycans [35]. MiRNAs are one of the most abundant nucleic acids in EVs, and they play a variety of roles in mediated cellular communication [36]. For example, miR-21 and 26a may play a role in cardiac hypertrophy remodeling, while miR-233 and 23a have been described as regulators of macrophage function in hypoxia and inflammation [37,38]. Other nucleic acid species carried by EVs include ribosomal RNA (rRNA), long non-coding RNA (lncRNA), transfer RNA (tRNA), and small nuclear RNA (snRNA), all of which have different influences on physiological processes [36].

In terms of proteins, CTS-EVs carry both proteins generalized to all EVs with regards to their biogenesis/release, as well as effector functions for their microenvironment. Among these effector proteins, transport proteins, such as annexins and Rab, are included [39]. Donor cells may release exosomes containing MHC Class I and II proteins as part of immune surveillance, while cancer cells may do the same for immune avoidance [40]. Cancer cell EVs can also release antiapoptotic factors, such as thioredoxine and peroxidase, for similar reasons [41]. Some CTS-EVs carry metabolic enzymes, including fatty acid synthase, G3P-dehydrogenase, ATPase, and more [39]. Other EVs have been shown to transport receptors such as tumor necrosis factor-alpha (TNF-α) receptors, transforming growth factor-beta (TGF-β) receptors, and transferring receptors [36]. Lipids not only have a structural role in CTS-EV membranes but also contribute to the regulation of release and communication. EVs are enriched in cholesterol, sphingomyelin, glycosphingolipids, phosphatidylserine, and other fatty acids [42]. Membrane lipids have been observed to interact directly with receptor molecules or act as second messenger molecules, thus allowing EVs to have multiple mechanisms to participate in cell signaling [43]. These bioactive lipids from EVs can be internalized into recipient cells, with an example study showing that accumulations of fatty acids internalized by exosomes create a prostaglandin-dependent biological response [36,44]. 

Finally, glycans are typically found as part of glycoproteins and glycolipids. These complex molecules are crucial for exosome biogenesis, release, uptake, and interaction with recipient cells [45]. The specific types of glycans present in exosomes can serve as biomarkers for certain diseases or conditions, and they can influence the biological functions of the exosomes [45,46].

### 2.3. CTS-EVs in Intercellular Communication

EVs have been observed to interact in cell signaling in two primary methods: direct interaction and membrane fusion. In direct interaction, a surface ligand on an EV binds with a target surface-bound receptor, which initiates a downstream cellular signaling cascade [31]. This can directly impact cellular functions and may additionally trigger the endocytosis of the EV. In the second method, EVs fuse with the target cells’ plasma membrane directly and release their contents into the cytoplasm [31]. Families of SNARE and Rab proteins modulate this fusion, ensuring that a cell only accepts certain EVs [47]. Lysosomes and other phagosomes break apart the lipid bilayer of the EV to expose the internal contents [48]. The various RNAs and proteins carried by the EV are released, which can affect gene expression and the modulation of intracellular signaling pathways. CTS-EVs exhibit a remarkable ability to target specific cell types with individualized effects, a feature that is crucial for their role in selective intercellular communication (Figure 1). Moreover, due to the sheer volume of EVs released, they play vital roles in maintaining the overall balance of homeostasis in the body. Examples of this will be outlined in the sections below.

## 3. CTS-EVs in Physiology and Pathology

### 3.1. Normal Physiological Function

In normal physiological processes, CTS-EVs play a fundamental role in maintaining normal physiological functions through their involvement in various aspects of cellular communication and homeostasis. EVs are capable of traversing various barriers within the body (including the blood–brain barrier between the bloodstream and brain parenchyma) to facilitate intercellular communication and transport biomolecules between different physiological compartments. The cell-specific nature of their cargo ensures that their actions are precise and targeted, thereby facilitating the fine-tuning of changes in response to tissue status [49]. 

CTS-EVs have emerged as crucial modulators of the immune system [50]. They possess the ability to convey signals that either activate or suppress immune responses, thereby playing a pivotal role in immune regulation. All immune cell types engaged in inflammation can produce EVs, and these EVs serve multiple functions in inflammatory processes [51,52]. EVs carry bioactive lipids, enzymes, and even molecules known as eicosanoids, which guide the movement of cells toward a specific location (chemotactic effects) [53]. Eicosanoids are signaling molecules that create chemical gradients in the body’s tissues. Cells, particularly immune cells, respond to these gradients and migrate toward the source of the eicosanoid signal. This chemotactic response is crucial in processes such as inflammation, where such signaling directs immune cells to sites of injury or infection. Through the transfer of these chemotactic agents, EVs facilitate critical interactions and responses in the body’s immune and healing processes. 

CTS-EVs are also shown to be pivotal in innate and adaptive immunity, including but not limited to inflammation, antigen presentation, and the maturation and activation of both B cells and T cells. Studies on exosomes isolated from human thymic epithelial cells (TECs), which are present in thymic tissue in both mice and humans, have been shown to carry tissue-restricted antigens (TRAs), such as myelin basic protein and desmoglein 3. This finding suggests that exosomes derived from thymic epithelia could play a crucial role in the thymocyte selection process. By disseminating self-antigens, TEC exosomes potentially enhance the exposure of maturing thymocytes to a wider range of TRAs, aiding in the critical processes of positive and negative selection within the thymus and thus the proper functioning of central tolerance mechanisms to prevent immune reactions [51,54]. CTS-EVs are also known to transport growth factors and signaling molecules that promote cell proliferation and differentiation, aiding in tissue regeneration and maintenance [55,56]. Dermal-related CTS-EVs, including those from dermal fibroblasts and keratinocytes, contribute to extracellular matrix (ECM) deposition and tissue remodeling [57,58]. Moreover, blood-derived EVs, such as those from platelet-rich plasma, umbilical cord blood, and endothelial cells, can regulate fibroblasts and contribute to angiogenesis in cutaneous tissue [59,60]. 

EVs are essential for the development and maintenance of the central nervous system (CNS) and are secreted from all types of neural cells, including neurons, astrocytes, microglia, and oligodendrocytes. These vesicles play critical roles in neuronal communication, synaptic function, and the regulation of neurogenesis [61,62,63]. Early research in this field revealed a correlation between enhanced neuronal activity and an uptick in the release of extracellular vesicles, a process thought to be crucial for the effective maintenance of synapses. CTS-EVs facilitate the exchange of proteins, miRNAs, and other signaling molecules between cells, contributing to the dynamic regulation of the neural environment both in the CNS and peripheral nerves, such as in the cornea [6,64]. This exchange helps maintain synaptic plasticity, which underlies learning and memory. Lopez et al. found that CTS-EVs from human neural rosettes and pluripotent stem cells (PSCs) play a critical role in human CNS development [65]. These EVs carry neuronal and glial cellular components necessary for brain development and play a role in influencing PSC morphology [66]. In addition to synaptic pruning, the regional development of the CNS is significantly influenced by intercellular communication. Sonic hedgehog (Shh) signaling, a key regulator of cortical development, exemplifies this. Tanaka and colleagues observed that the embryonic fluid flow in developing mice causes the release of EVs in a leftward direction. These EVs, carrying Shh, contribute to the lateralization of the CNS, highlighting their role in developmental processes [67].

CTS-EVs are vital in regulating metabolic processes, including insulin sensitivity and lipid metabolism [68,69]. Studies revealed that EVs from adipose tissues modulate insulin signaling pathways, impacting glucose homeostasis and peripheral insulin sensitivity [70,71]. These vesicles are also important in lipid metabolism, transporting lipids and lipid-related enzymes and receptors, primarily in the liver and adipose tissues, where they influence lipid storage and breakdown [72]. Additionally, EVs facilitate metabolic communication between organs like the liver, adipose tissue, and skeletal muscle, which is essential for maintaining energy balance and metabolic stability [73]. These insights underscore the importance of EVs in metabolic regulation and their growing relevance in metabolic disease research and treatment. Additionally, in current cell-based and tissue engineering approaches, including co-culture systems, there is growing evidence that the differentiation of various cell types, including neurons, is remarkably improved with the presence of specific surrounding cells [74]. Neuronal differentiation benefits from the presence of cells like meningeal cells or astrocytes [66,75]. However, there is also evidence indicating that the secretion of soluble factors alone, without the direct inclusion of the supporting cells, may be sufficient to augment the differentiation and maturation processes. These secreted factors, including EVs, can improve key aspects of cellular differentiation by delivering a range of bioactive molecules, such as proteins, lipids, and various types of RNA [76,77]. 

In addition to the CTS-EVs’ roles in normal physiological processes, CTS-EVs can either exacerbate disease progression or function as potent (or impotent) modulators in various disease states. Table 1 systematically catalogs the current understanding of CTS-EVs’ involvement across a diverse array of diseases and disorders, categorizing them according to disease type for ease of reference. Subsequent sections of this review follow the sectional numbering provided in Table 1.

### 3.2. Cancer

Within the context of cancer, EVs from both tumor and non-tumor sources have been shown to play a role in tumor migration, immunosuppression, and angiogenesis. Cancer CTS-EVs can contain metalloproteinases (MMPs), which degrade ECM, allowing for the local invasion of tumor cells [78]. Furthermore, EVs from cancer cells can cause fibroblast dedifferentiation into cancer-associated fibroblasts, which can lead to metastasis [79]. It has been shown that tumor-derived EVs can carry immunosuppressive cargo that prevents immune cells from attacking circulating tumor cells. One such “cargo” is the protein programmed death ligand 1 (PD-L1). When this receptor binds with T cells, it may induce immunosuppression [80]. Natural killer (NK) cells are also affected by a similar mechanism, where EVs expressing ligands of NKG2D lead to the immunosuppression of NK cells [81]. Tumor-derived EVs can also stimulate macrophages to release the factors that promote angiogenesis, further increasing the tumor’s vascular permeability [82]. EVs from cancer stem cells (CSCs) are linked to an increase in epithelial-mesenchymal transition (EMT) status, which can enhance the aggressiveness of the cancer. For instance, miR-21, which has been involved in many antiapoptotic pathways, is a common component of tumor-derived EVs [83]. In vitro studies also demonstrated that when lung cancer cells are treated with CSC-derived EVs, the lung cancer cells become more invasive [84].

Due to the dramatic impact of EVs on cancer progression, it is important to study the composition of cancer-associated EVs to help diagnose and treat cancer. In certain cancers, including melanoma, glioblastoma, prostate cancer, breast cancer, and more, there is a higher concentration of all circulating EVs in plasma for cancer patients versus healthy donors [85,86,87,88,89]. However, the true cause of the increased EV count is still controversial. There is some evidence that blood serum EV levels drop after the primary tumor mass is surgically removed [85,90]. In addition, there is evidence that cancer can be detected not only by counting the number of EVs in a sample but also by identifying specific biomarkers within these EVs found in liquid biopsies, like blood and urine [85,91]. Tumor-specific EV biomarkers may provide a more precise method for cancer diagnosis than tissue biopsies. Several studies reported that analyzing the contents of EVs by way of liquid biopsies can be a successful alternative to more invasive tissue biopsies [85,91,92,93]. Some of the biomarkers found in EVs include miR-1246 (breast cancer), PSA (prostate cancer), and upregulated lncRNA (liver cancer) [85]. EVs that are known to target cancer can be modified to incorporate strategies to combat cancer [94]. For example, lung cancer cell-derived EVs can be loaded with an oncolytic adenovirus combined with chemotherapeutic drugs [95]. EVs derived from immune cells have also been shown to be helpful for cancer therapy [96]. For example, Wang et al. found that neutrophilic-derived exosomes from an inflammatory tumor microenvironment can be used to deliver DOX for targeted glioma therapy [96]. EVs that target cancer cells can also be used to deliver gene therapy—namely the CRISPR/Cas9 system—and can be effective for normally hard-to-transfect cells and cancer cells [97]. M1 macrophage-derived EVs combined with PD-L1, a checkpoint inhibitor, also enhance cancer therapy [96]. The diagnostic and therapeutic use of EVs has been explored for many different cancers, including breast cancer, lung cancer, lung cancer, prostate cancer, leukemia, ovarian cancer, and glioblastoma. Examples are given below.

For breast cancer diagnosis, specifically, there are several potential biomarkers. Connexin-46 (Cx-46), a protein that is important for gap junction channels [98], is shown to be present in high concentrations in inflammatory breast cancer (IBC)-derived EVs [99]. Inflammatory carcinoma of the breast is an often missed diagnosis and is confused with mastitis, so having a liquid biopsy-based cancer detection could be transformational [94,100]. To detect whether breast cancer is estrogen receptor-based or not, 241 unique serum-derived EV (sEV) proteins were profiled [93]. sEV protein biomarkers can also be used for breast cancer prognosis; higher levels of developmental endothelial locus-1 (Del-1) from plasma-derived sEVs were used as prognostic indicators [101,102]. Clinical trials have been conducted to use EVs in diagnostics, including a study from 2014 to use tumor-derived exosomes as a diagnostic and prognostic marker [103]. To target triple-negative breast cancer, engineered EVs were loaded with Adriamycin, a chemotherapeutic medication, and a modified version of miR-159 [104]. EVs tend to act as fertilizers for the tissue from which they were sourced—thus leading to a very large range of options in terms of cancer therapy [82]. A handful of circulating nucleic acids can be used for the diagnosis and prognosis of lung cancer [105]. One such example is a clinical trial using ctDNA from pulmonary exosomes to identify benign vs. malignant pulmonary nodules [106]. EVs can also be used to detect lung cancer treatment effectiveness, as EVs extracted from a non-small cell lung cancer cell line that was treated with gefitinib were found to reduce cisplatin’s antitumor effectiveness [107,108]. Cisplatin usually increases the autophagy of cells; however, it was found to have an antagonistic relationship with gefitinib [107]. Several EV-based biomarkers have been found for prostate cancer, including a test for TMPRSS2:ERG in urine-derived EVs, which mimics the results of prostate biopsies. There are additional EV-based genes that can be used as biomarkers, including PCA3, ERG, BIRC5, and TMPRRS2 [109]. Several drug-resistance biomarkers were also found for prostate cancer prognosis [91]. For detecting castrate-resistant prostate cancer, a form of prostate cancer that persists after androgen depletion therapy, the presence of androgen receptor AR-Variant 7 in plasma-derived EVs was of special interest [110]. In one of the more common forms of leukemia, acute lymphoblastic leukemia (ALL), leukemic cells released EVs containing miR-43a-5p to the bone marrow [111]—which caused the inhibition of osteogenesis, leading to leukemia progression [112]. In acute myeloid leukemia (AML), a panel of miRNAs from serum EVs can be used to distinguish leukemic mice from controls [110,113]. For the treatment of chronic lymphocytic leukemia (CLL), a study used B cells that were targeted by EVs tagged with a viral envelope protein (gp350) and transferred gp350 and CD40L to the patients’ cells, leading to an anti-CLL immune response [114]. Other proteins can be transferred to patients’ cells, such as pp65 [115] or chemotherapeutics like fludarabine [114,116]. In ovarian cancer, there are a handful of potential protein biomarkers for diagnosis. A strong CD24 presence also coincided with epCAM, a known marker of cancer, in EVs from an ovarian cancer cell line, which confirmed that the strong expression of CD24 was indeed tumor-derived [117]. Keseru et al. claimed that the mitochondrial DNA (mtDNA) copy number found in plasma-derived EVs was significantly greater in all stages (except stage 1) of ovarian cancer [118], which could prove to be a valuable tool in detecting early-stage ovarian cancer [119]. A specially made EV engineering system named Immune Derived Exosome Mimetics (IDEM) containing doxorubicin was able to be produced in large quantities and was effective in drug delivery for the treatment of ovarian cancer cells [120]. There are other EV-based therapeutic approaches for ovarian cancer, such as ExoSTING, that are being developed [119]. EVs in urine have shown promise for enhancing glioblastoma identification in an assay called “liquid gold”. A study concluded that EV concentration in urine can be correlated to glioblastoma prognosis, prompting further investigations into the diagnostic capabilities of EVs [121]. Current clinical trials are ongoing, including one demonstrating the detection by next-generation sequencing of molecular abnormalities present in exosomes from glioblastomas to non-invasively identify tumor subtypes [122].

### 3.3. Neurological Disorders 

Traditionally, communication within the brain’s neural network has been attributed primarily to synaptic vesicles moving from pre- to post-synaptic neurons [123]. However, recent studies showed that CTS-EVs released from neurons, astrocytes, oligodendrocytes, and microglia represent an emerging mode of intraneuronal communication [124]. These CTS-EVs transfer proteins and genomic materials between neurons, playing a role in both normal neuronal functions and the propagation of pathological proteins in neurodegenerative diseases. CTS-EVs can offer a non-invasive approach to diagnosing and monitoring neurological disorders, as well as a potential therapeutic tool for treating these conditions. Because EVs can pass through the blood–brain barrier, if loaded with therapeutics, they can also transport therapeutics through the blood–brain barrier [124]. Alzheimer’s disease (AD) is characterized by memory, cognitive functioning, and behavioral impairments because of abnormal neurofibrillary tangles and neurotic plaques. AD is known to induce β-amyloid plaque sheaths in the neural membrane. Neuron-derived CTS-EVs are known to transport amyloid-β peptides and tau proteins, which are key in the disease’s progression [125]. These vesicles assist in spreading these neurotoxic agents, leading to synaptic dysfunction and widespread neuronal damage [126]. Due to the nanoparticle size of EVs, biolabeling the particles with microRNA markers allows their location to be visualized [127]. Specifically, miR-9-5p, miR-598, and miR-125b are shown to be highly stable and proliferative, which allows for identification in patients [127,128,129,130]. You et al. identified a specific protein, integrin β1, that was found to be elevated in CTS-EVs from astrocytes associated with brain β-amyloid and tau load [131]. 

Parkinson’s disease (PD) is a neurodegenerative disease characterized by tremors, rigidity, and akinesia. In PD, the dopaminergic neurons of the substantia nigra are lost or reduced [23,132]. EVs can propagate PD pathogenesis by interacting and transporting α-synuclein, a protein whose misfolding and aggregation are hallmark features of the disease. It has been demonstrated that plasma EVs containing α-synuclein can be used as a minimally invasive biomarker of PD [133]. Additionally, EVs can be shown to deliver dopamine across the blood–brain barrier, interacting with appropriate receptors and reducing the negative effects of dopamine deficiency [134]. Multiple sclerosis (MS), a central nervous system autoimmune disease, is characterized by inflammation and immune dysfunction caused by demyelination of the CNS. CTS-EVs originating from distinct cell types are implicated in MS pathology, encapsulating specific biomarkers, including myelin-derived proteins such as myelin basic protein (MBP), proteolipid protein (PLP), and myelin oligodendrocyte glycoprotein (MOG). These biomarkers are indicative of the autoimmune response targeting myelin components in MS [135]. The investigation of CTS-EVs allows for a meticulous dissection of the molecular nuances, offering a targeted understanding of immune dysregulation in MS. Moreover, the bioactive cargo within these EVs, consisting of anti-inflammatory cytokines and neurotrophic factors, presents tailored opportunities for therapeutic modulation, aiming at specific cellular pathways contributing to MS pathogenesis [135,136]. Oligodendrocyte-derived EVs (OD-EVs) have proved to be useful markers of cerebrovascular accidents (strokes). Oligodendrocytes provide an established role in remyelination in post-ischemic injury and influence neuronal survival [137,138]. Fröhlich et al. examined the effects of OD-EVs in an oligodendrocyte/neuron co-culture stressed with ischemic injury and saw that cells treated with OD-EVs had significantly higher metabolic activity compared with control cells [139]. OD-EVs appeared to contain cargo, including enzymes, such as catalase and superoxide dismutase 1, that provided antioxidant benefits [139]. Broadly, knowledge of EVs derived from OD-EVs and EVs in the context of stroke remains limited, with more extensive research efforts needed [137]. 

Traumatic brain injury (TBI) is difficult to assess and identify because no known biomarkers are readily accessible, and imaging techniques lack the specificity to visualize specific affected brain injuries. EVs can pass through the blood–brain barrier, which can enable them to be engineered to deliver molecular tracers, thus potentially illuminating injured areas within the brain without the need for high-resolution scanning neurotechnology. EVs are also implicated with neuronal–glial cell communication and reduced neuroinflammation when derived from the neuronal cells of the CNS [140]. 

Mental disorders encompass a wide range of neurological diseases. Depression, for example, is thought to be caused by neuroinflammation and hormone imbalance, but the effect of CTS-EVs on depression has not been thoroughly investigated. A study showed that exosomes derived from patients with the major depressive disorder, when injected into healthy mice, caused the mice to exhibit depressive-like behavior. This study found that the major difference in these exosomes compared with exosomes from healthy persons was hsa-miR-139-5p upregulation [141]. Another study investigated the expression levels of microRNA-26a (miR-26a) in EVs in the hippocampus of depressed rats [142]. The study uncovered that the expression of miR-26a was significantly reduced in the hippocampus of rats with depression. This discovery positions miR-26a as a potential diagnostic marker for depression, highlighting its importance in identifying the condition. In amyotrophic lateral sclerosis (ALS), EVs are associated with TDP-43 fragments and SOD1 mutations. While TDP-43 pathology correlates with disease severity, suggesting a potential use as a biomarker, associations with SOD1 mutations remain unclear, casting doubt on their utility for biomarker monitoring [143]. Misfolded proteins associated with EVs represent a promising avenue for biomarker discovery in prion diseases, as they may allow the detection of aberrant isoforms. Studies in EV isolation techniques have allowed the identification of intercellular communication mechanisms through CTS-EVs that contribute to infection in prion disease [144,145]. Transport of misfolded proteins also extends to Huntington’s disease, where the mutant huntingtin protein is exported via vesicles. The detection of mutant huntingtin in EVs in plasma provides a less invasive way to track disease progression [146]. Changes in the viscoelasticity of EVs induced by mutant huntingtin offer a new approach for the rapid identification of misfolded proteins. 

### 3.4. Ophthalmic Conditions

CTS-EVs have been studied for their potential role as biomarkers or in the broader context of diagnostics in various diseases of the eye. One example is the use of CTS-EVs derived from the retinal pigment epithelium (RPE) as potential biomarkers for age-related macular degeneration (AMD). Klingeborn et al. discuss the isolation of retinal EV biomarkers from blood using targeted immunocapture, which represents a promising approach for early AMD biomarker development [147]. The RPE, being a critical location for disease-associated changes in AMD, sheds EVs that enter the systemic circulation and thus could be harnessed for blood-based biomarker discovery. Further research, as mentioned by Kang et al., explored EV proteins in the aqueous humor as potential biomarkers for neovascular AMD, comparing the protein profiles in the aqueous humor from individuals with the disease to unaffected individuals [148]. These studies suggest that EVs and, in particular, exosomes could serve as valuable biomarkers for diagnosing and monitoring the progression of AMD. Furthermore, EVs derived from the urine and retinal tissue of diabetic retinopathy (DR) patients have shown distinct protein profiles when compared to those from healthy individuals. This study found that junction plakoglobin (JUP), a protein associated with cell–cell junctions, was present in EVs from DR patients but not in the controls [149]. This suggests that JUP could potentially serve as a biomarker for DR, offering opportunities for earlier diagnosis and better prognosis. Another study explored the impact of plasma-derived EVs on angiogenesis and the progression of proliferative diabetic retinopathy (PDR), with a focus on the role of microRNA-30b. This research suggests that the mechanisms influenced by plasma-derived EVs may regulate angiogenesis in PDR via a mechanism dependent on miR-30b [150].

In the context of glaucoma, recent studies have been investigating the role of CTS-EVs in intraocular pressure regulation and optic nerve damage [151]. Research demonstrates that CTS-EVs play a role in the pathology of glaucoma, particularly in the regulation of intraocular pressure and signaling within the ocular system. By comparing the proteome of EVs isolated from trabecular meshwork (TM) cells of both glaucomatous and non-glaucomatous human donors, it was found that EVs from glaucomatous TM cells had a different protein profile, with a reduced abundance of key extracellular matrix proteins, suggesting a mechanism for ECM regulation in glaucoma and potentially revealing targets for future therapies [152]. 

Uveal melanoma (UM) is the most common intraocular cancer, which is also known to spread to the liver, causing incurable metastases. It was found by Piquet et al. that uveal melanoma cells, when compared to healthy melanocytes, release significantly more EVs that also exhibit hepatotropism [153]. These EVs also induced angiogenesis and fibrosis on hepatic stellate cells and endothelial cells, which was not found with EVs that came from melanocytes. Melanoma EVs have been found to increase metastatic UM cell attachment to the liver through reprogrammed hepatic stellate cells by modifying the extracellular matrix [153]. 

With regards to Dry Eye Syndrome (DED), Andrew et al. explored the association between inflammatory-related tear film microRNAs (miRNAs)-derived EVs and EVs in non-Sjogren’s syndrome DED [154]. Tear samples from DED and non-DED individuals were analyzed for EVs and miRNAs. The findings revealed EVs in both groups’ tear films, with RNA-Seq identifying 126 differentially expressed miRNAs. Nine of these miRNAs, linked to inflammation, were significantly upregulated in DED patients. The research suggested that miRNAs in tear film EVs are involved in DED’s inflammatory pathways. Another study investigated the role of EVs in the epithelial-to-mesenchymal transition (EMT) of lens epithelial cells under oxidative stress, a process related to posterior capsule opacification (PCO), a common complication after cataract surgery [155]. The study found that EVs and reactive oxygen species (ROS) were involved in this transition. The results suggested that EVs participate in ROS-induced lens EMT, indicating that EVs could be a potential target for treating PCO. 

In relation to uveitis, a study conducted in 2023 focused on profiling CTS-EVs in uveitis and scleritis to discover biomarkers and explore their mechanisms. This comprehensive analysis involved proteomics of small and large EV subpopulations from patients with various uveitis types and posterior scleritis. The study identified over 3000 proteins and found that EV proteomic profiles correlated more with disease than plasma proteomic profiles [156]. It highlighted potential pathogenic mechanisms and identified biomarker panels for these diseases, providing insights for diagnosis and potential therapeutic targets. 

The human cornea and its optical transparency are crucial for vision [157]. It has been proposed that EVs are actively secreted in corneal epithelial defects to enable communication between stroma and epithelium, suppressing scar formation and maintaining surface clarity [5,157]. Both corneal fibroblast and MSC-derived EVs have been specifically identified to be able to regulate the inflammatory response and modulate nerve regeneration through the transport of signaling proteins such as TGF-β [158,159,160,161]. Another study found corneal epithelial EVs to have the ability to influence the differentiation of conjunctival and corneal epithelial cells, further participating in injury remodeling [158,162]. In another study, the differential impacts of exosomes derived from non-diabetic (N) and diabetic (DM) human limbal epithelial cells (LECs) on limbal stromal cells (LSCs) were analyzed, uncovering critical insights into corneal health and diabetic complications. A key finding of this study was that exosomes from N LECs significantly enhance wound healing and cell proliferation in LSCs, unlike those from DM LECs. This effect was attributed to the distinct miRNA and protein compositions of the exosomes, suggesting a crucial role of cellular communication mechanisms in diabetic corneal conditions [163]. Another study from the same group highlighted the role of miR-146a in corneal wound healing, showing it is upregulated in the diabetic limbus and reduces pro-inflammatory responses in LECs [164]. The overexpression of miR-146a decreased inflammation-related genes and mediators, particularly in diabetic conditions. This suggests miR-146a’s potential as a therapeutic target for managing diabetic corneal complications. Research concerning corneal CTS-EVs and their applications are still in the early stages but show great potential for impacting disease management in the future.

### 3.5. Infectious Disease

In the context of infectious diseases, EVs play complex roles. They participate in immune system processes and have been implicated in both the progression and protection against diseases. In infections, EVs can carry viral and host components, modulating the immune response. Proteomic analysis of EVs from patients with infections like SARS-CoV-2 has provided insights into their role in disease progression and potential as biomarkers. These findings point to a critical role for EVs in understanding and managing infectious diseases, with implications for developing new therapeutic and diagnostic tools. Elettra et al. conducted a proteomic analysis of circulating exosomes in patients with SARS-CoV-2 to explore the host response to the virus [165]. It revealed that these exosomes carry molecules related to immune response, inflammation, coagulation, and complement pathways, which are crucial in COVID-19 pathogenesis and organ damage. Notably, the research identified potential biomarkers for disease severity and diagnostic accuracy. It also discovered SARS-CoV-2 RNA within exosomes, suggesting a possible mechanism for viral spread [165]. This study provided significant insights into the role of exosomes in inflammation, coagulation, and immune modulation during COVID-19 infection and highlighted their potential as biomarkers. 

Furthermore, the research conducted by Masatoshi et al. on Hepatitis B Virus (HBV) illustrated that apart from producing infectious virions, HBV also generates various non-infectious particles, including EVs [166]. These EVs, particularly exosomes, play a crucial role in cell-to-cell communication. They are taken up by monocytes/macrophages, primarily leading to the upregulation of PD-L1, which is involved in immune response suppression. Importantly, the study showed that the characteristics and effects of these EVs are modified by nucleotide reverse transcriptase inhibitors (NRTIs), suggesting a significant role in HBV pathogenesis and the clinical response in patients treated with NRTIs. 

In the context of HIV, EVs have both pro-viral and antiviral effects [167]. For example, HIV-1-infected EVs have been demonstrated to carry gp120, a key envelope protein of HIV, which enhances viral entry in lymphoid tissues [168]. They additionally increase disease progression through the use of T-cell immunoglobulin and mucin domain-containing 4 (TIM4), which can facilitate the entry of HIV-1 into host cells. EVs containing surface TIM4 bind with the virus, and the trafficking of HIV-1 to immune cells is hastened, resulting in increased infectivity and disease progression [169]. With regards to antiviral properties from the host, EVs released by CD8 T cells can carry ligands that have anti-HIV properties [170]. In addition, it has been reported that EVs trigger innate immunity to restrict HIV. When Toll-like receptor 3 is activated in human brain endothelial cells, they release EVs containing interferons and other antiviral factors that can prevent HIV infection [171]. 

### 3.6. Autoimmune Diseases

CTS-EVs have become a focal point in the scientific exploration of autoimmune conditions. Generally, immunosuppression is the pathway followed to treat many autoimmune diseases (ADs), though, after some time, the complexity of the human immune system counters its efforts [172]. Many of today’s steroidal and non-steroidal therapies have dull results due to their inability to offer highly specific effects. EVs, on the other hand, have highly configurable packages and can deliver in a targeted fashion [172,173]. 

Rheumatoid Arthritis is an autoimmune disease leading to a chronic inflammatory state primarily affecting the synovial tissues encapsulating joints. The underlying cause is multifactorial, involving genetic predispositions, including human leukocyte antigen (HLA) alleles and environmental triggers [174]. EVs from regulatory T cells (Tregs), characterized by anti-inflammatory microRNAs, may offer a protective influence by suppressing immune responses. Conversely, EVs released by activated immune cells, carrying pro-inflammatory cytokines, could worsen joint inflammation. Biomarkers such as miR-155 and miR-146a in EVs from Tregs may signify immunoregulatory potential, while EVs containing citrullinated proteins or inflammatory mediators may mark heightened autoimmunity [174,175,176]. Additionally, EVs from mesenchymal stem cells may harbor reparative factors like transforming growth factor-β (TGF-β), presenting a positive influence, while those carrying matrix metalloproteinases might contribute to tissue degradation [175,176,177,178]. 

The exploration of CTS-EVs in systemic lupus erythematosus (SLE) reveals a unique molecular signature indicative of the underlying immune dysregulation. These EVs carry specific biomarkers, including antinuclear antibodies (ANAs), anti-double-stranded DNA antibodies (anti-dsDNA), and anti-Smith antibodies, pinpointing cellular targets within SLE [179]. Additionally, cytokines carried by CTS-EVs, such as interleukin-6 (IL-6) and tumor necrosis factor-alpha (TNF-α), contribute to the inflammatory environment associated with SLE [180]. Focusing on biomarkers from implicated cell types, such as B cells and dendritic cells, offers a targeted understanding of SLE pathogenesis. Leveraging these specific biomarkers holds promise for precise diagnostic assays and informs potential therapeutic strategies aimed at modulating the identified molecular pathways contributing to immune dysregulation in SLE [179,180].

In Sjogren’s syndrome, a disorder affecting the exocrine glands, the scrutiny of CTS-EVs as diagnostic and therapeutic entities is anchored in their distinct molecular composition [172,181]. Originating from salivary and lacrimal gland cells, these EVs encapsulate hallmark biomarkers, including anti-Ro (SSA) and anti-La (SSB) antibodies, indicative of autoimmune responses [181]. Focusing on CTS-EVs enables the targeted exploration of molecular intricacies tied to glandular dysfunction and local inflammation. Additionally, the bioactive cargo within these EVs, containing cytokines like interleukin-17 (IL-17) and interferon-gamma (IFN-γ), provides insight into the pro-inflammatory environment characterizing Sjogren’s syndrome [181,182]. This nuanced investigation facilitates a refined understanding of the disease and lays the groundwork for personalized diagnostics and therapeutic interventions.

In Irritable Bowel Syndrome (IBS), a complex gastrointestinal disorder, the exploration of CTS-EVs as diagnostic and therapeutic agents involves a meticulous analysis of diverse biomarkers encapsulated within these vesicles [183]. Derived from various cell types in the gastrointestinal tract, these EVs carry biomarkers such as serotonin, tryptase, and inflammatory cytokines, shedding light on altered neurotransmission, mast cell activation, and immune dysregulation inherent to IBS pathology [183,184,185]. The specificity of investigating CTS-EVs enables a targeted examination of the molecular signatures associated with distinct cellular contributors to IBS. Furthermore, the bioactive cargo within these EVs, encompassing neuropeptides, pro-inflammatory mediators, and additional markers like calprotectin and fecal lactoferrin, offers insights into the neuroimmune interactions contributing to symptoms such as altered gut motility and visceral hypersensitivity in IBS [186]. This intricate investigation of cell type-specific EVs not only deepens our understanding of IBS at the molecular level but also opens avenues for personalized diagnostics and therapeutic strategies by leveraging a comprehensive array of biomarkers. In Myasthenia Gravis (MG), a neuromuscular junction autoimmune disorder, the investigation of CTS-EVs unveils a nuanced molecular landscape. These EVs, originating from diverse neuromuscular junction cell types, encapsulate biomarkers like acetylcholine receptor (AChR) antibodies and muscle-specific kinase (MuSK), offering insights into the immunological responses underlying MG [187,188]. The targeted study of CTS-EVs enables a focused examination of molecular signatures associated with neuromuscular dysfunction. Additionally, the bioactive cargo within these EVs, comprising miRNAs and cytokines, provides a comprehensive view of the immune modulation and inflammatory processes in MG [188]. The emerging research on CTS-EVs in MG holds promise for innovative diagnostics and personalized therapeutic strategies, enhancing our understanding and management of this complex autoimmune disorder.

### 3.7. Cardiovascular Diseases

Nearly one-third of deaths globally are caused by cardiovascular diseases (CVDs), which constitute a major cause of mortality [189]. According to a number of recent studies, cells associated with the cardiovascular system, including cardiomyocytes, endotheliocytes, fibroblasts, platelets, smooth muscle cells (SMCs), leucocytes, monocytes, and macrophages, can generate EVs [190]. Under physiological conditions, EVs have vital roles in maintaining proper heart structure and function. Conversely, in pathological circumstances, EVs alter their composition and aid in the formation of CVDs [191]. It has been demonstrated that the main sources of EVs in individuals with stable coronary artery disease (CAD) are endotheliocytes and platelets. miR-92a-3p is packaged into endothelial EVs under atherosclerotic circumstances, which, in turn, controls angiogenesis in recipient endotheliocytes through a process that is reliant on thrombospondin-1 [192]. Through the miR-939-mediated nitric oxide signaling pathway, coronary serum EVs isolated from individuals with myocardial ischemia (MI) induce angiogenesis [193]. In individuals with stable CAD, the increased expression of miR-126 and miR-199a in EVs but not plasma has been linked to a decrease in major adverse cardiovascular events [194]. In MI, in response to hypoxia and inflammation, cardiomyocytes release EVs enriched in miRNAs linked to endothelium proliferation and differentiation, such as miR-143 and miR-222 [195]. miR-15-b could also play a mechanistic role in the dysregulation of cardiac EVs and during the release from cardiac EVs following ischemic damage [196]. MI has been linked to a circulating EV protein network that also plays a role in inflammatory response [197]. Furthermore, EVs are essential for the remodeling of the myocardium following an MI, indicating a great deal of potential for therapies involving the upregulation or downregulation of miR-containing EVs [191]. 

In individuals with atrial fibrillation, miR-21-3p has been detected in EVs and plasma with links to abnormal cardiac tissue expansion. It has been demonstrated that miR-199a-3p in extracellular matrix-derived EVs stimulates the formation of heart tissue and influences atrial electrical activity by upregulating Gata-binding 4 (Gata4) acetylation and suppressing the expression of homeodomain-only protein X (HOPX) [198]. In the development of peripartum cardiomyopathy, endotheliocyte-secreted EVs loaded with miR-146a function as both biomarkers and messengers [199]. A 16 kDa N-terminal prolactin fragment (16 K PRL) stimulates endotheliocytes to produce more miR-146a and facilitates the production of EVs laden with miR-146a. Maladaptive hypertrophy is caused by miR-146a, which also decreases the expression of small ubiquitin-like protein modifier 1 (SUMO1) and causes cardiac dysfunction [200]. In summary, CTS-EVs play a crucial role in both the maintenance of cardiovascular health and the progression of cardiovascular diseases. CTS-EVs, originating from various cell types within the cardiovascular system, are key in regulating heart function and structure under normal conditions. Conversely, in diseased states, they contribute to the development and progression of CVDs through mechanisms involving specific microRNAs and proteins. This dual role of EVs underscores their importance in cardiovascular research and highlights their potential as targets for therapeutic intervention in heart-related diseases.

### 3.8. Hematologic Disorders

In physiological states, CTS-EVs secreted from blood cells, especially platelet-derived EVs, have garnered considerable scientific interest. Their abundance in the bloodstream and pivotal roles in processes such as wound healing, angiogenesis, thrombosis, and atherosclerosis make them a significant focus of study [201,202,203,204]. The isolation of these EVs, which requires further study, could aid the understanding and treatment of a variety of clotting disorders and thrombocytopenias [205]. Red blood cell-derived EVs (RBC-EVs) have also attracted attention due to their potential role in erythropoiesis, iron homeostasis, and hemolysis. Expelled RBC-EVs contain hemoglobin, which can be taken up by macrophages and recycled, as well as other proteins and lipids that may affect cellular functioning [206]. Harnessing these EVs could aid in replacing hemoglobin in anemias such as sickle cell anemia. White blood cell-derived EVs (WBC-EVs), including those from neutrophils, monocytes, and lymphocytes, have been implicated in various immune responses, such as phagocytosis, antigen presentation, and cytokine production [52]. WBC-EVs can also transfer antigens and adjuvants to dendritic cells, leading to the activation of T cells and the initiation of adaptive immunity in those who have immunodeficiencies [207].

### 3.9. Reproductive Disorders

It has been reported that EVs participate in the conception and implantation processes by cross-talk with sperm and embryo cells, which leads to normal and healthy births. EV-mediated pathways in reproduction can be explored as an alternative to invasive amniocentesis and chorionic villi sampling [208]. In polycystic ovarian syndrome (PCOS), Sang et al. identified miRNAs within the CTS-EVs of human follicular fluids that decreased in PCOS compared with healthy patients [209]. These miRNAs were previously identified to be associated with key roles in the etiology of PCOS in a previous genome. Additionally, CTS-EVs from endometrial stromal cells from women with endometriosis compared with healthy patients demonstrated different profiles of EV miRNA cargo [208,210]. Among these miRNAs, miR21 is already known for its role in angiogenesis, which is associated with the pathology of endometriosis.

### 3.10. Metabolic Diseases

EVs have been implicated in the regulation of metabolic homeostasis. A meta-analysis conducted by Li et al. revealed elevated levels of a variety a CTS-EVs originating from platelets, endothelial cells, and monocytes in individuals with type 2 diabetes with cargo including annexin V, CD31, CD104, and CD62E [211,212]. Kobayashi et al. found that circulating CTS-EV numbers were higher in men and even more so in individuals with impaired glucose tolerance. These EVs, particularly from hypertrophic adipocytes, correlated strongly with insulin resistance and triglyceride levels. Additionally, EVs were associated with β-cell function and metabolic parameters, including lipid and glucose metabolism. The presence of specific markers on some EVs suggests their origin from adipocytes and hepatocytes, reinforcing their potential as metabolic biomarkers and their role in monitoring human metabolic health [213]. Increased CTS-EV quantity additionally correlated with hypertension and increased body mass index (BMI). In another study, patients with obesity (BMI 30+ kg/m^2^) were found to exhibit a plasma EV concentration approximately ten times higher than individuals of normal weight [214,215,216,217]. These EVs consist of about 20% exosomes and 80% microvesicles. This rise in plasma EV levels has also been observed in overweight patients (BMI 25–30 kg/m^2^) [215]. Notably, subjecting patients to a reduced-calorie diet, a regimen of diet and exercise, or weight reduction following sleeve gastrectomy surgery led to a decline in their plasma EV levels [214,215]. Campello et al. found that although there was a decrease in these levels after post-sleeve gastrectomy, they did not match the levels observed in healthy individuals [218].

### 3.11. Liver Disease

Recently, it has been documented that CTS-EVs regulate liver cell proliferation, differentiation, and metabolism as they carry and transfer several liver-specific proteins and miRNAs [219,220]. CTS-EVs harbor different types of RNAs that can regulate functions in other hepatic stellate cells (HSCs), which are crucial in liver fibrosis in conditions like nonalcoholic fatty liver disease (NAFLD), with their activation being a key factor. Davide et al. explored the role of CTS-EVs released by fat-laden hepatocytes in triggering this activation. The CTS-EVs containing miRNAs like miR-128-3p were found to be internalized by HSCs, leading to their activation, and increased pro-fibrogenic gene expression, proliferation, and migration. These effects were primarily due to the suppression of PPAR-γ expression in HSCs by the miRNAs in EVs. These findings suggested that hepatocyte-derived EVs play a significant role in liver fibrosis, offering potential new anti-fibrotic targets for NAFLD and other related diseases [221]. In another study focusing on insulin resistance, HepG2 liver-derived cells were used to investigate the cleavage of the insulin receptor (IR), highlighting the significant role of exosomes [222]. The research found that in these liver-derived cells, elevated glucose levels accelerate the cleavage of the IR. This process is facilitated by calpain 2, which is secreted into the extracellular space in association with exosomes, leading to the cleavage of the IRβ subunit [222]. This finding was crucial, as it links liver metabolism to impaired insulin signaling, emphasizing the role of exosomes in metabolic pathways associated with insulin resistance and the potential impact of treatments like metformin. In hepatocellular carcinoma, a cancer with only a 17% 5-year survival rate, EVs have shown oncogenic behavior by transporting miR-21 and miR-223, which participate in tumor growth and metastasis [219,223,224,225]. Finding specific CTS-EV targets for hepatocellular carcinoma can provide potential targets for diagnosis, prevention, and future treatment.

### 3.12. Renal Diseases

An increasing amount of research has been conducted to establish the role of CTS-EVs in kidney physiology and pathology. Chronic kidney disease (CKD) follows a slow, progressive course with biopsy as the gold standard diagnostic tool to determine etiology. The evaluation of urinary small EVs may serve as a “liquid” biopsy [226,227]. As an example, Barutta et al. found urinary extracellular vesicles from patients with diabetic nephropathy, showing an enrichment of miR-130a and miR-145 and a reduction in miR-155 and miR-424 [228]. Another study found increased levels of miR-200b with a decrease in miR-29c in urinary CTS-EVs isolated from CKD patients compared with healthy individuals [229]. CTS-EVs additionally have therapeutic potential, as pre-clinical studies have utilized MSC-EVs to spur kidney regeneration [226]. Eirin et al. treated localized tubular cells, a model of renal artery stenosis with MSC-EVs carrying IL-10, and found improved renal inflammation and oxygenation [230]. Renal cell carcinoma (RCC)-derived CTS-EVs have also been found to influence the tumor microenvironment for the promotion of cancer progression [226]. Several markers, including metalloproteinase 9, nephrilysin, ceruloplasmin, podocalyxin, and more, have been identified to have different profiles within urinary EVs in RCC compared with healthy patients [231]. This niche tumor microenvironment also provides a crucial step in potential metastasis formation. Long non-coding RNA (lncRNA) have been implicated, in particular, with Jin et al. finding that RCC CTS-EVs promote cancer cell migration and metastasis by carrying a lncRNA called lung adenocarcinoma transcript 1 (MALAT1), which is also associated with other cancer metastases [232,233]. 

### 3.13. Respiratory Diseases

CTS-EVs have been observed to have roles in many respiratory diseases. In acute lung injury (ARI) and acute respiratory distress syndrome, mouse models have shown an increase in EVs filled with inflammatory molecules that were sent to alveolar macrophages after the initial insult [234,235]. The EV profiles of both infectious and non-infectious lung injury showed an increase in CTS-EVs containing cytokines, such as IL-6, TNF-α, TLR2, TLR6, and IL-10, that were dispersed to alveolar macrophages [235]. The detection of these CTS-EVs could help predict the likelihood of progression in potential ARDS cases. Asthma is a chronic process featuring chronic inflammation and airway hyperresponsiveness. Lung CTS-EVs containing miRNA, such as miR-145, have been identified as key mediators in smooth muscle function for asthma-induced bronchoconstriction [234,236]. Though asthma is considered incurable with current medication, CTS-EVs could be used as a potential therapeutic option. These EVs have been found to suppress pro-inflammatory responses and reduce pulmonary fibrosis-related remodeling in inflammatory lung disease animal models, with the added benefit of a respected safety profile [237]. COPD is another chronic inflammatory lung process resulting in exacerbations that can lead to poor oxygenation and bacterial infection. CTS-EVs have been thought to be implicated in the pathophysiology, with a study finding an increase in exosome levels in COPD patients, with those exosomes being observed to mediate leukotriene conversion in affected tissues [8,234,238,239]. Specifically, increased concentrations of EV-miR-21 were found in the bronchial epithelial cells of COPD patients [234,240]. EV-miR-21 is hypothesized to directly control the autophagy functions of myofibroblasts, with autophagy dysregulation being involved in COPD progression [240,241]. The specificity of EV-miR-21 creates the possibility of a clinically relevant biomarker for COPD.

### 3.14. Dermatological Diseases

Recent studies have indicated that EVs play key immunomodulatory roles in skin disorders such as psoriasis, atopic dermatitis, melanoma, and more. Psoriasis is a chronic inflammatory dermatologic disease driven by keratinocytes and the infiltration of immune cells [242]. Studies have demonstrated that platelet and endothelial cell CTS-EVs are increased in psoriasis patients and are additionally correlated with disease severity [243,244,245,246]. These CTS-EVs were found to be reduced by anti-TNF-α treatment, a mainstay of severe psoriasis treatment [243,247]. One of the most common skin inflammatory skin disorders is atopic dermatitis (AD), or eczema, and these patients are highly suspectable to bacterial skin infections, which in turn, drive disease exacerbation [243,248]. Zhou et al. observed that S. epidermidis EVs significantly attenuated inflammatory cell inflammation and IgE levels in AD mice models [249]. Others have found that S. aureus EVs could exacerbate AD by carrying mediators that increase the expression of inflammatory cytokines, such as IL-6 [250]. Thus, these EVs could represent potential therapeutic targets when managing AD exacerbations. Skin cancer is one of the most important diagnoses in the field of dermatology, with squamous cell carcinoma (SCC) of the skin being one such example. Li et al. recently reported that cutaneous SCC cells present an increased number of EVs compared with normal epidermal keratinocytes [251]. Moreover, tumor xenograft analysis via RNA sequencing revealed that the impaired EV production of cutaneous SCC cells impaired phenotype transformation and inhibited tumor growth [251]. These findings could serve as both a diagnostic biomarker and a therapeutic target for SCC management in the future.

### 3.15. Challenges and Emerging Trends

In the field of using CTS-EVs as diagnostic tools and to understand their roles in normal physiology, several challenges are intertwined with emerging trends. One of the primary challenges is achieving high specificity and sensitivity in diagnosis because accurately identifying EVs from specific cells amid a diverse population of body fluids is complex. Additionally, the lack of standardized protocols for EV isolation and analysis introduces variability, complicating their interpretation as biomarkers. A further significant hurdle is the limited understanding of the roles of these EVs in normal physiological processes, making it difficult to differentiate between physiological and pathological EV signatures. Moreover, current technological limitations in detailed EV analysis present challenges, especially given their small size and complex cargo.

However, these challenges are being met with innovative responses. Advanced isolation techniques, such as microfluidics, bead-capture flow cytometry, magnetic-based bead absorption strategies, and nanoparticle-based methods, among others, are being developed to improve the purity and yield of specific EVs. The use of high-throughput sequencing and proteomics is enhancing the characterization of EVs, providing deeper insights into their diagnostic potential. Artificial Intelligence and machine learning are increasingly used to analyze complex EV datasets, aiding in the identification of disease-specific diagnostic patterns. The integration of cell type-specific EVs into personalized medicine is gaining traction, promising more tailored diagnostic and prognostic strategies. Additionally, there is a growing focus on functional studies of EVs in normal physiological contexts. Such studies are crucial for distinguishing the roles of CTS-EVs in health versus disease. Together, these advancements are paving the way for the effective utilization of CTS-EVs in both understanding normal physiological processes and developing novel diagnostic approaches. 

Additionally, while EVs have emerged as crucial elements in the realm of therapeutic delivery, the challenge of achieving efficient and reproducible loading of EVs with therapeutic agents remains a critical barrier. In addressing this issue, studies have introduced a methodology involving the modification of EVs through their fusion with liposomes containing both membrane and soluble cargoes [252]. This innovative approach entails the triggering of EV and liposome fusion via polyethylene glycol (PEG), resulting in the creation of smart biosynthetic hybrid vectors. The developed method exhibits versatility and efficiency in augmenting EVs with exogenous lipophilic or hydrophilic compounds while preserving their intrinsic content and biological properties. Notably, these hybrid EVs have demonstrated a substantial enhancement in cellular delivery efficiency for a chemotherapeutic compound, outperforming both the free drug and the drug-loaded liposome precursor by a factor of 3–4 [253]. This perspective holds significant promise for enhancing the characteristics of numerous drug delivery formulations, especially those that have already obtained regulatory approval and market authorization [254].

## 4. Conclusions

As covered in this review, research in the area of CTS-EVs has demonstrated their great importance in cellular signaling and homeostasis through a variety of physiological systems. CTS-EVs are also implicated in propagating diseased states, such as cancer, neurodegenerative disorders, infectious organisms, and more. A thorough identification of the involved CTS-EVs allows for a better understanding of the pathophysiology behind a variety of human diseases. Moreover, these CTS-EVs have promising potential as diagnostic markers, with their unique molecular signatures and carried content that circulates in a variety of body fluids, allowing for non-invasive testing. Understanding the specific proliferative or non-proliferative effect EV cargo may have on a disease state additionally opens up the possibility of targeted therapeutics, with EVs themselves serving as fitting non-invasive delivery mechanisms.

## Figures and Tables

**Figure 1 ijms-25-02730-f001:**
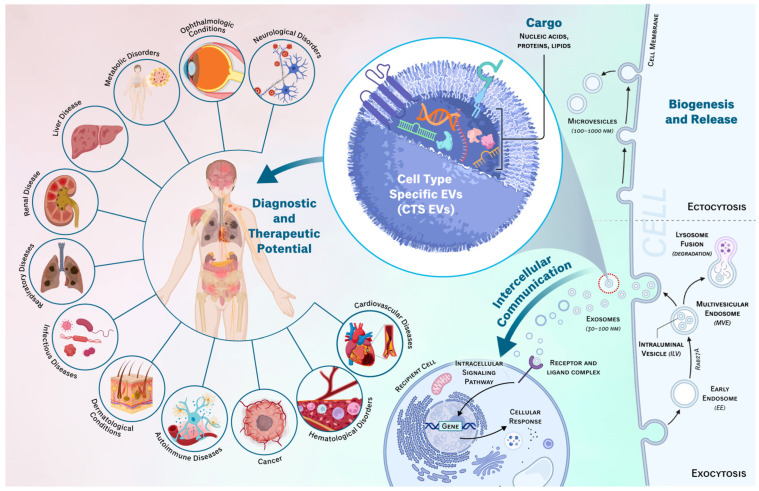
Schematic of CTS-EVs and their role in different physiological and pathological processes. From right to left, the process of microvesicle and exosome biogenesis and exocytosis from their cell of origin is detailed. CTS-EV responses have a variety of diagnostic and therapeutic implications in a variety of disease types, as shown here. This image was created using BioRender and subsequently modified by our group.

**Table 1 ijms-25-02730-t001:** Overview of Diseases and Disorders Associated with CTS-Derived EVs.

	Disease Type	Specific Diseases	References
3.2	Cancer	Breast, Lung, Prostate, Leukemia, Ovarian, Glioblastoma, Pancreatic, Colorectal, Bladder, Oropharyngeal, Gastric	[75,76,77,78,79,80,81,82,83,84,85,86,87,88,89,90,91,92,93,94,95,96,97,98,99,100,101,102,103,104,105,106,107,108,109,110,111,112,113,114,115,116,117,118,119]
3.3	Neurological Disorders	Alzheimer’s, Parkinson’s, Multiple Sclerosis, Stroke, Epilepsy, Depression, Traumatic Brain Injury (TBI), Multiple Sclerosis, Huntington’s, ALS, Prion disease	[120,121,122,123,124,125,126,127,128,129,130,131,132,133,134,135,136,137,138,139,140,141,142,143,144]
3.4	Ophthalmologic Conditions	Age-Related Macular Degeneration (AMD), Diabetic Retinopathy, Glaucoma, Uveitis, Retinal Dystrophies, Dry Eye Disease, Corneal Trauma, Posterior Capsular Opacification, Ocular Melanoma	[1,5,6,7,145,146,147,148,149,150,151,152,153,154,155,156,157,158,159,160,161,162]
3.5	Infectious Diseases	Human Immunodeficiency Virus (HIV), Hepatitis Viruses, COVID-19, Tuberculosis	[50,163,164,165,166,167,168,169]
3.6	Autoimmune Diseases	Rheumatoid Arthritis (RA), Systemic Lupus Erythematosus (SLE), Sjogren’s Syndrome, Inflammatory Bowel Diseases (IBD), Myasthenia Gravis, Autoimmune Thyroid Diseases, Celiac Disease, Systemic Sclerosis (Scleroderma), Antiphospholipid Syndrome	[170,171,172,173,174,175,176,177,178,179,180,181,182,183,184,185,186]
3.7	Cardiovascular diseases	Myocardial Infarction, Coronary Artery Disease, Atrial Fibrillation, Cardiomyopathy	[187,188,189,190,191,192,193,194,195,196,197,198]
3.8	Hematological Disorders	Thrombosis, Sickle cell anemia, Hemophilia, Anemia, Thrombocytopenia, Immunodeficiency	[50,199,200,201,202,203,204,205]
3.9	Reproductive Disorders	Endometriosis, Infertility, Genetic Testing, Polycystic Ovarian Syndrome (PCOS)	[206,207,208]
3.10	Metabolic Disorders	Diabetes, Metabolic Syndrome, Obesity	[209,210,211,212,213,214,215,216]
3.11	Liver Disease	Hepatocellular Carcinoma, Liver Fibrosis, Cirrhosis	[217,218,219,220,221,222,223]
3.12	Renal Diseases	Chronic Kidney Disease, Renal Artery Stenosis, Renal Cell Carcinoma	[224,225,226,227,228,229,230,231]
3.13	Respiratory Diseases	Acute Lung Injury (ARI), Acute Respiratory Distress Syndrome (ARDS), Chronic Obstructive Pulmonary Disease (COPD), Asthma	[8,232,233,234,235,236,237,238,239]
3.14	Dermatological Conditions	Psoriasis, Atopic Dermatitis, Skin Cancer	[240,241,242,243,244,245,246,247,248,249]

## Data Availability

Not applicable.

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
