# Peer review of "Cell Type-Specific Extracellular Vesicles and Their Impact on Health and Disease"

_ijms, 2024, doi:10.3390/ijms25052730_

Round 1

Reviewer 1 Report

Comments and Suggestions for Authors

A review article entitled "Cell Type Specific Extracellular Vesicles and Their Impact on Health and Disease"

This work is well-organised and could be accepted pending minor revision as follows:

1. The Figure-1 is of a low resolution and should be improved. 

2. The authors should discuss the therapeutic application.

3. The authors should prepare a table summarizing the various types of Extracellular Vesicles and their impacts in various type of disease. The current Table-2 is not detail enough.

Comments on the Quality of English Language

Acceptable.

Author Response

Thank you very much for taking the time to review this manuscript, please find responses to your points below along with an updated document reflecting said changes.

  1. The resolution of the image has been updated, please refer to the image in the attached revised paper and the figure legend in figure 1 is also updated.

  1. Thank you for your comment, we recognize the only brief mention of therapeutic applications in the paper, as in our review we found therapeutic applications primarily focused on MSC-derived EVs as opposed to CTS-EVs. We have added additional mentions of specific MSC-EV based therapeutic studies as well as a few in the CTS-EV applications, such as ophthalmology. The relevant sections have been highlighted and in the attached paper (section introduction and 3.2 and 3.14) and also here for your reference and also mentioned here, particularly in the introduction, diseases section, and challenges/emerging trends.

“The field of EVs is gaining great attention because of EVs’ unique role in intercellular communication. Predominant research in this field has concentrated on therapeutic applications of mesenchymal stem cell derived EVs (MSC-EVs), which have demonstrated a significant effect across diverse diseases including [14,15], neurological disorders [3], cardiovascular disorders [16], respiratory disorders [17], metabolic disorders [18], ophthalmic conditions [1,19] and various cancers [20,21]. For example, utilizing their ability to cross the blood-brain barrier, MSC-EVs have been employed for the encapsulation of curcumin and a magnetic resonance imaging contrast agent in the treatment of Parkinson’s disease (PD), with the miRNA within these exosomes additionally demonstrating a preventive effect on neuronal death and a reduction in PD symptoms [22,23].

 EVs that are known to target cancer can be modified to incorporate strategies to combat cancer [94]. For example, lung cancer cell-derived EVs can be loaded with an oncolytic adenovirus combined with chemotherapeutic drugs [95]. EVs derived from immune cells have also been shown to be helpful for cancer therapy [96]. For example, Wang et al. found that neutrophilic derived exosomes from an inflammatory tumor microenvironment can be used to delivery DOX for targeted glioma therapy [96]. EVs that target cancer cells can also be used to deliver gene therapy—namely the CRISPR/Cas9 system—and can be effective for normally hard-to-transfect cells and cancer cells [97]. M1 macrophage-derived EVs combined with PD-L1, a checkpoint inhibitor, also enhanced cancer therapy [98].

Additionally, while EVs have emerged as crucial elements in the realm of therapeutic delivery, the challenge of achieving efficient and reproducible loading of EVs with therapeutic agents remains a critical barrier. In addressing this issue, studies have introduced a methodology involving the modification of EVs through their fusion with liposomes containing both membrane and soluble cargoes [255]. This innovative approach entails the triggering of EV and liposome fusion via polyethylene glycol (PEG), resulting in the creation of smart biosynthetic hybrid vectors. The developed method exhibits versatility and efficiency in augmenting EVs with exogenous lipophilic or hydrophilic compounds while preserving their intrinsic content and biological properties. Notably, these hybrid EVs have demonstrated a substantial enhancement in cellular delivery efficiency for a chemotherapeutic compound, outperforming both the free drug and the drug-loaded liposome precursor by a factor of 3 – 4 [256]. This perspective holds significant promise for enhancing the characteristics of numerous drug delivery formulations, especially those that have already obtained regulatory approval and market authorization."

  1. Upon literature review, we have found that a vast majority of EV based applications in the study of mechanisms, diagnostics, and therapeutics refer to only one type of EVs, exosomes isolated through centrifugation techniques, so there as not as many applications for the other types of extracellular vesicles (such as microvesicles and apoptotic bodies). This was confirmed in nearly all the studies we have in our review of specific disease related papers. To address the reviewer’s suggestion we added a sentence to clarify this in section 2.1. It is important to note that a vast majority of EV based applications in the study of mechanisms, diagnostics, and therapeutics refer to cell type specific exosomes in particular, isolated through centrifugation techniques, so there as not as many applications for the other types of extracellular vesicles, such as microvesicles and apoptotic bodies. The original purpose of our existing table was intended as more of a reference for readers with further interest in the specific diseases, with the references we have cited being organized for easy access to additional reading. Overviews of these diseases are detailed below in the specific disease sections. If you had feedback as to what components should specifically be introduced into our table, it would be appreciated.

  1. All references are updated.

Reviewer 2 Report

Comments and Suggestions for Authors

The research in the area of CTS-EVs has demonstrated their great importance in cellular signaling and homeostasis through a variety of physiological systems.

CTS-EVs are also implicated in propagating diseased states, such as cancer, neurodegenerative disorders, infectious organisms and more.

Thorough identification of the involved CTS-EVs is allowing for better understanding the pathophysiology behind a variety of human diseases.

Moreover, these CTS-EVs have promising potential as diagnostic markers, with their unique molecular signatures and carried content that circulate in a variety of body fluids allowing for non-invasive testing.

Understanding the specific proliferative or non-proliferative effect EV cargo may have on a disease state additionally opens up the possibility of targeted therapeutics, with EVs themselves serving as fitting non-invasive delivery mechanisms.

Author Response

Thank you for your kind comments and taking the time to review our manuscript, we appreciate the feedback of having a clear, defined message as well as having appropriate references selected. We have implemented these additional changes from the other reviewer’s advice, with changes highlighted for your reference.

Round 2

Reviewer 1 Report

Comments and Suggestions for Authors

The revision is acceptable for publication.

Comments on the Quality of English Language

Acceptable